# The CoDiNOS trial protocol: an international randomised controlled trial of intravenous sildenafil versus inhaled nitric oxide for the treatment of pulmonary hypertension in neonates with congenital diaphragmatic hernia

Suzan Cochius-den Otter ,[1] Thomas Schaible,[2] Anne Greenough ,[3] Arno van Heijst,[4] Neil Patel,[5] Karel Allegaert,[6] Joost van Rosmalen,[7] Dick Tibboel,[1] on behalf of the CDH EURO Consortium

For numbered affiliations see end of article.

**Correspondence to**
Dr Suzan Cochius-den Otter; s.denotter@erasmusmc.nl

## ABSTRACT

**Introduction** Congenital diaphragmatic hernia (CDH) is a developmental defect of the diaphragm that impairs normal lung development, causing pulmonary hypertension (PH). PH in CDH newborns is the main determinant for morbidity and mortality. Different therapies are still mainly based on 'trial and error'. Inhaled nitric oxide (iNO) is often the drug of first choice. However, iNO does not seem to improve mortality. Intravenous sildenafil has reduced mortality in newborns with PH without CDH, but prospective data in CDH patients are lacking.

**Methods and analysis** In an open label, multicentre, international randomised controlled trial in Europe, Canada and Australia, 330 newborns with CDH and PH are recruited over a 4-year period (2018–2022). Patients are randomised for intravenous sildenafil or iNO. Sildenafil is given in a loading dose of 0.4 mg/kg in 3 hours; followed by continuous infusion of 1.6 mg/kg/day, iNO is dosed at 20 ppm. Primary outcome is absence of PH on day 14 without pulmonary vasodilator therapy and/or absence of death within the first 28 days of life. Secondary outcome measures include clinical and echocardiographic markers of PH in the first year of life. We hypothesise that sildenafil gives a 25% reduction in the primary outcome from 68% to 48% on day 14, for which a sample size of 330 patients is needed. An intention-to-treat analysis will be performed. A p-value (two-sided) <0.05 is considered significant in all analyses.

**Ethics and dissemination** Ethics approval has been granted by the ethics committee in Rotterdam (MEC-2017-324) and the central Committee on Research Involving Human Subjects (NL60229.078.17) in the Netherlands. The principles of the Declaration of Helsinki, the Medical Research Involving Human Subjects Act and the national rules and regulations on personal data protection will be used. Parental informed consent will be obtained.

**Trial registration number** NTR6982; Pre-results.

## Strengths and limitations of this study

► The Congenital Diaphragmatic hernia Nitric Oxide versus Sildenafil (CoDiNOS) trial is the first randomised controlled multicentre trial to evaluate the effect of intravenous sildenafil and compare with inhaled nitric oxide (iNO) on pulmonary hypertension in newborns with congenital diaphragmatic hernia (CDH).

► Treatment allocation is not blinded in the trial. This is not feasible because of variability in iNO equipment and gas mixtures use. Instead, the researchers who analyse the echocardiography to evaluate pulmonary hypertension (PH) will be blinded to the treatment.

► The primary outcome, PH, will be measured using echocardiography instead of just clinical parameters often used in newborns.

► There is no non-intervention group, as it is common practice in the centres of the CDH EURO Consortium to give iNO; hence, it is considered unethical to withhold treatment for one group.

► Long-term follow-up of 12 months will give more insight in the course of PH in infants.

## INTRODUCTION

Congenital diaphragmatic hernia (CDH) is a severe developmental defect of the diaphragm with an incidence of approximately 1 in 3000 live births and a mortality of 27%.[1] Because of this defect, the abdominal organs herniate into the chest causing pulmonary hypoplasia and abnormal pulmonary vasculature growth, resulting in pulmonary hypertension (PH).[2] In adults and children, PH is defined as mean pulmonary artery pressure (mPAP) exceeding 25 mm Hg with a pulmonary capillary wedge pressure of minimal 15 mm Hg.[3]

The normal pulmonary vascular transition of the neonate takes around 2 months to achieve these low values of mPAP. During fetal

life, there is high resistance in the pulmonary circulation which results in most of the blood flow to bypass the lungs through the ductus arteriosus and oval foramen. Immediately after birth, the pulmonary vascular resistance drops and the blood flow to the lungs significantly increases.[4] In contrast, the pulmonary vascular resistance often does not drop adequately in children with CDH due to a decreased vascular bed associated with lung hypoplasia, and an altered development of the pulmonary vasculature with excessive muscularisation of the arterioles, with increased thickness of the arterial media and adventitia. Although the presence of lung hypoplasia can be predicted with prenatal parameters, reliable predictors for PH in CDH patients are lacking.[5] The incidence of PH in CDH patients is 68%–79% and causes considerable morbidity and mortality.[1 2 6] Therapy in newborns with PH, such as inhaled nitric oxide (iNO) and sildenafil, has improved outcomes in general. However, trials in infants with CDH are sparse.

iNO diffuses rapidly across the alveolar-capillary membrane into the smooth muscle of pulmonary vessels to activate soluble guanylate cyclase. This enzyme mediates many of the biological effects of iNO, and is responsible for the conversion of guanosine triphosphate (GTP) to cyclic guanosine monophosphate (cGMP). The increase of intracellular cGMP relaxes smooth muscles via several mechanisms. iNO also causes bronchodilation and has anti-inflammatory and anti-proliferative effects.[7] In term and near term infants with persistent pulmonary hypertension of the newborn (PPHN), iNO decreases the median duration of mechanical ventilation and reduces the need for extracorporeal membrane oxygenation (ECMO). However, in the two available randomised controlled trials (RCT) with a small number of patients with CDH, mortality did not improve and more ECMO treatment was needed despite short-term improvement of oxygenation in some treated patients.[8 9] In the centres of the CDH EURO Consortium, iNO is standard of care in infants with CDH and PH although the positive pharmacodynamic effects in these infants are less convincing than in infants with PPHN.[6 10] The pathophysiological mechanism of this difference is not understood. In resource poor settings iNO is often unavailable. In the search to find another treatment option, trials to evaluate the effect of sildenafil in newborns with PPHN have been conducted.[11]

Sildenafil is a selective phosphodiesterase type 5 (PDE5) inhibitor. PDE5 is an enzyme that specifically degrades cGMP. Sildenafil inhibits PDE5, increasing cGMP and nitric oxide-mediated vasodilatation of the smooth muscles in arteries. Only five RCTs have been performed in newborns, all non-CDH patients with PPHN. Four of these studies showed a decrease in oxygenation index (OI) and mortality in a setting where iNO was not available, while one trial showed no additional benefit of sildenafil when added to iNO.[11] Although sildenafil is increasingly used in CDH patients, only retrospective data are available.[12] A decrease in pulmonary vascular resistance index and an increase in cardiac output were found in a small group of oral sildenafil-treated infants with CDH refractory to iNO.[13] Intravenous sildenafil improved OI and reversed the right-to-left shunt ratio over the PDA, but it also increased the need for inotropic support.[14 15] However, its effect on outcome is unknown.

We hypothesise that intravenous sildenafil is superior to iNO. iNO is the therapy of first choice in most centres despite the lack of evidence, and sildenafil is the most promising drug for the treatment of PH in CDH patients and is increasingly being used.[6 12 16] However, no studies have been performed comparing iNO with intravenous sildenafil in newborns with CDH and PH or PH alone. Based on the current knowledge, there is equipoise for both treatment modalities.

## METHODS AND ANALYSIS
### Design
The Congenital Diaphragmatic hernia Nitric Oxide versus Sildenafil (CoDiNOS) trial is a prospective, multicentre, international RCT conducted in high volume paediatric surgical centres in Europe, Canada and Australia. The members of the CDH Euro Consortium participating in the trial are listed in the online supplementary appendix.

### Objectives
The primary objective of the study is to determine whether the incidence of PH is lower in CDH patients treated with intravenous sildenafil than in patients treated with iNO, with the primary outcome defined as the absence of PH on echocardiography on day 14 without pulmonary vasodilator therapy and without treatment failure and/or death within the first 28 days after birth. PH is defined as systolic pulmonary arterial pressure >2/3 systolic systemic pressure and/or right ventricular (RV) dilatation/septal displacement and RV dysfunction +/−left ventricular dysfunction.

The secondary outcomes are:
1. Change in OI after 12 and 24 hours of therapy.
2. Overall mortality.
3. The incidence of treatment failure which is defined as:
   - inability to maintain preductal saturations above 85% (±7 kPa or 52 mm Hg) or postductal saturations above 70% (±5.3 kPa or 40 mm Hg).
   - and/or increase in $CO_2$ >70 mm Hg (9.3 kPa) despite optimisation of ventilator management.
   - and/or inadequate oxygen delivery with metabolic acidosis defined as lactate ≥5 mmol/L and pH <7.15 and/or hypotension resistant to fluid therapy and adequate inotropic support resulting in a urine output <0.5 mL/kg/hour.
   - and/or lactate ≥5 mmol/L and pH <7.15.
   - and/or OI consistently ≥40.
4. Time on intervention drug, defined as intervention drug free days after initiation of the intervention, calculated on day 14.
5. Need for ECMO.
6. Ventilator free days on day 28.

7. The use of drugs for PH treatment during the hospital admission.
8. The use of pulmonary and/or cardiac medication at discharge and its total duration of administration.
9. Short-term and long-term PH on echocardiography at 24 hours, 28 days/discharge and 6 and 12 months.
10. The incidence of chronic lung disease.
11. The development of neurological abnormalities evaluated with ultrasound of the brain before the start of the trial, after surgery and before discharge.
12. The external validation of the sildenafil pharmacokinetic - pharmacodynamic (PKPD) model for the pharmacokinetics and the pharmacodynamic effects of sildenafil.

Safety outcomes include adverse events due to the study drugs and the vasoactive-inotropic support (VIS) score.

## Patients

Infants diagnosed with CDH who have PH in the first week after birth are eligible for the trial if born at or after a gestational age of 34 weeks. The diagnosis of PH is defined as at least two of the following four criteria: (1) systolic pulmonary arterial pressure >2/3 systolic systemic pressure estimated by echocardiography, (2) RV dilatation/septal displacement, RV dysfunction +/−left ventricular dysfunction, (3) pre-post ductal $SpO_2$ difference >10%, (4) OI >20. Exclusion criteria are a severe chromosomal anomaly which may imply a decision to stop or not to start life-saving medical treatment, severe cardiac anomaly expected to need corrective surgery in the first 60 days of life, renal anomalies associated with oligohydramnios, severe orthopaedic and skeletal deformities, which are likely to influence thoracic, and/or lung development and severe anomalies of the central nervous system. Patients who are born in another centre and transported with iNO are also excluded from the trial. Patients who received fetal interventions (trachea balloon placement) are not excluded.

Following antenatal diagnosis, the parents are counselled and informed about the study by the clinician or research coordinator. Also, they receive a patient information letter and an informed consent form . If the patient is not born in a participating centre or the diagnosis of CDH was not known, parents are counselled after the diagnosis of CDH and are informed about the study. Also, they receive written information and an informed consent form. This informed consent form contains consent for the trial and for collection of data and material for future research.

For the development of the protocol the SPIRIT reporting guidelines have been used.[17] This publication is based on protocol V.4, 13 June 2018.

## Patient and public involvement

Patients and the public were not involved in the development of the trial protocol. However, CDH UK Sparks, as a parent organisation, has assessed and commented on the protocol and has provided start-up funding as also mentioned in the funding statement. This organisation is and will be regularly informed on progress and results of the trial.

## Study procedures

### Baseline assessment

Antenatal ultrasound data about the characteristics of the CDH are collected. These data include the observed/expected lung-head ratio, position of the liver and stomach and the amniotic fluid index. An MRI or an ultrasound is performed depending on local experience and possibilities. If an MRI is performed, the observed/expected fetal lung volume will be calculated. Also data on prenatal interventions are collected. In all mothers, a planned vaginal or caesarean delivery is pursued.

### Randomisation, intervention and blinding

Participants will be randomised using ALEA, which is an online, central randomisation service (https://www.alea-clinical.eu). Allocation concealment will be ensured, as the service will not release the randomisation code until the patient has been recruited into the trial, which takes place after all baseline characteristics have been added. ALEA randomises the patient with a computer-generated randomisation list, made by the independent statistician of the Data Safety and Monitoring Board. Blocked randomisation, with variable block sizes and stratification by centre, is used to achieve equal distribution of the two interventions among the participants.

Postnatally, infants are treated according to a standardised protocol for patients with CDH, which is implemented in all participating centres. This protocol was developed with the available evidence and consensus between the participating centres and was updated in June 2016.[10 16] If the patient is diagnosed with PH in the first week of life, the patient will be allocated to one of the two study drugs (figure 1). iNO is provided by a tank connected to a ventilator. Different devices are used in different centres. Some centres use integrated systems, making it impossible to disconnect the iNO tank and replace it with another gas to facilitate a blinded intervention. Therefore, the study is open label. iNO is given with a starting dose of 20 ppm, which is the maximum dose.[18 19] Sildenafil is given intravenously, using a loading dose of 0.4 mg/kg in 3 hours, followed by continuous infusion of 1.6 mg/kg/day.[20 21] To wean the study drugs a standard protocol is followed (figure 2). The allocated drugs will be restarted as per protocol if criteria for its use are met again before the age of 14 days. To further standardise care, an inotropic support flow chart is included in the study protocol (figure 3). After day 14 treatment of PH will be at the discretion of the local medical team and the study drug can be changed to, for instance, sildenafil orally. The use of bosentan, milrinone and prostin next to the study treatment is allowed. The use of bosentan as add on therapy is allowed and is considered as PH treatment on day 14. The intervention will be prematurely stopped when the patient meets one

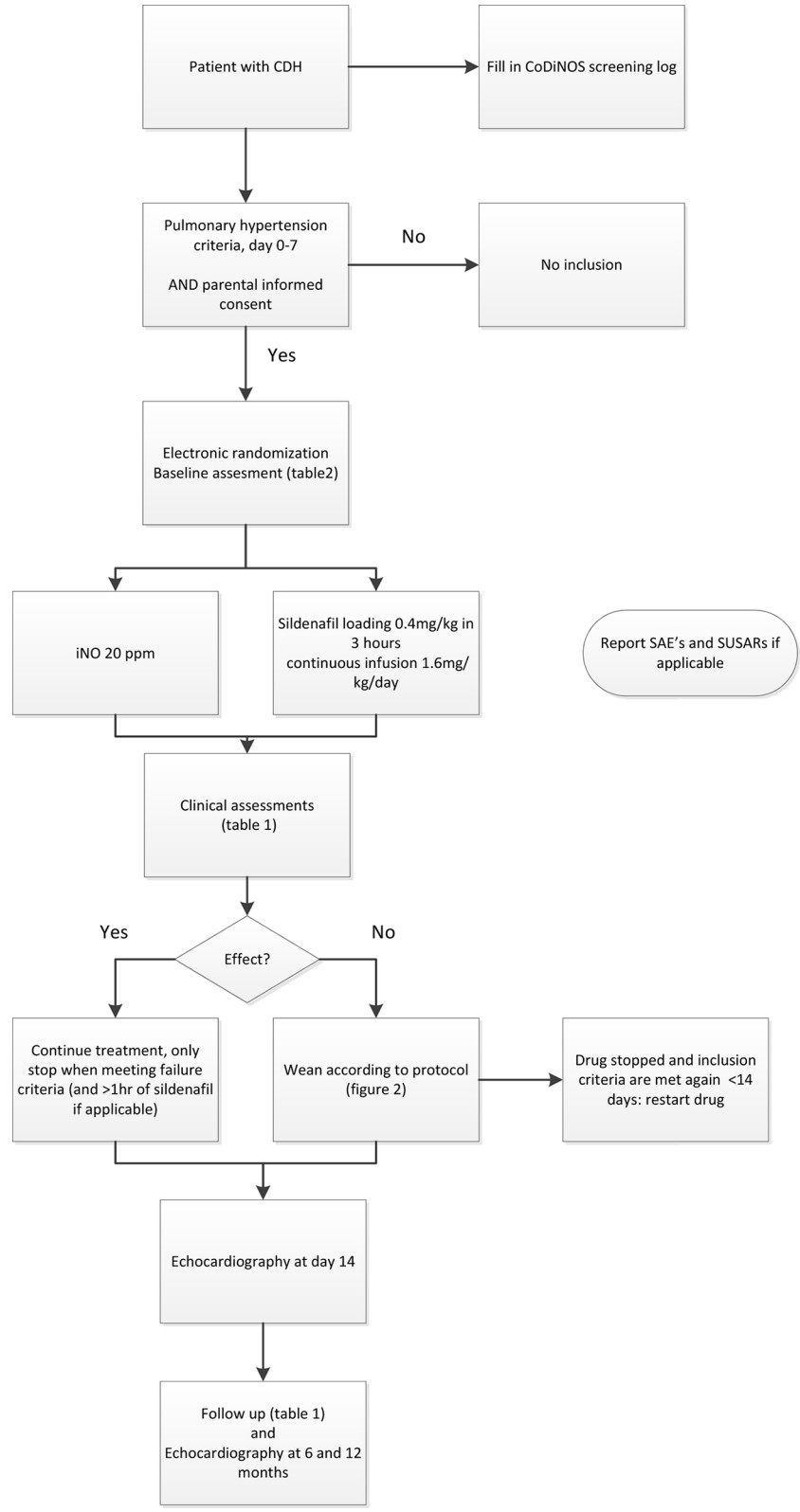

**Figure 1** Trial flow chart. Flow chart showing the steps of the trial, from birth until 12 months. CoDiNOS, Congenital Diaphragmatic hernia Nitric Oxide versus Sildenafil;CDH, congenital diaphragmatic hernia; SAE, serious adverse event; SUSAR, suspected unexpected serious adverse reaction.

or more of the defined failure criteria, described in point three of the secondary outcomes. Further treatment will then be at the discretion of the medical team and will be according to the standardised protocol.[16] iNO and sildenafil can both be given outside the study protocol. An ECMO procedure may then be started in centres where ECMO is available. Data of all patients are used in the intention-to-treat analysis.

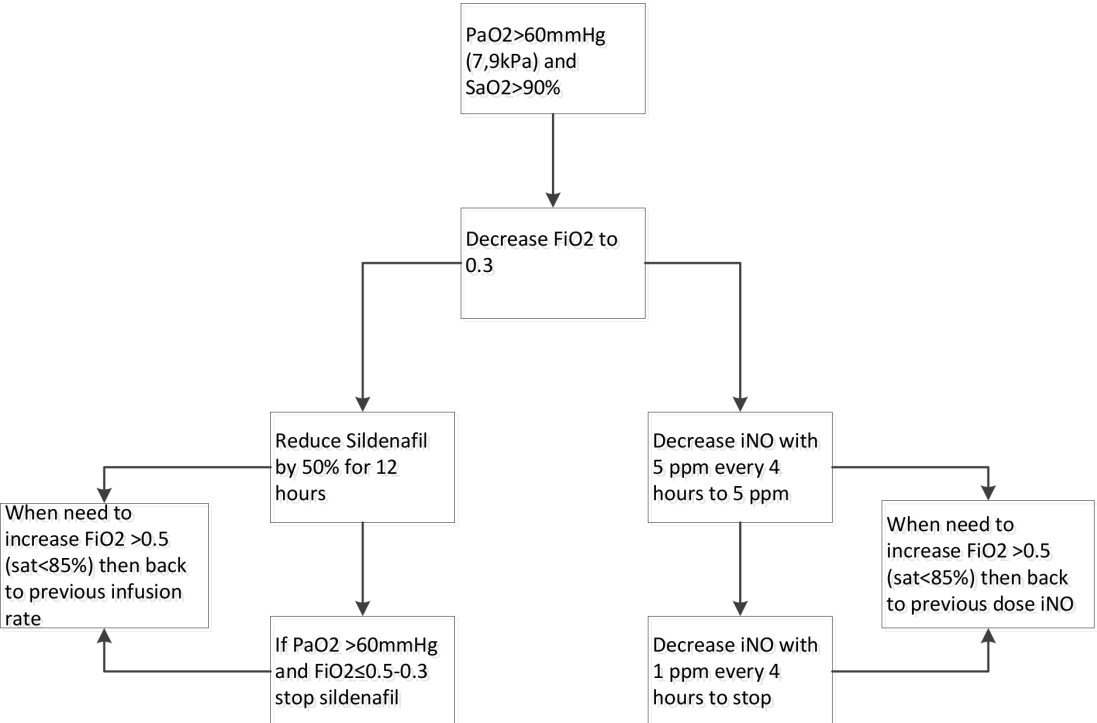

**Figure 2** Protocol to wean study drug. Flow chart showing the protocol to wean off inhaled nitric oxide or intravenous sildenafil. INO, inhaled nitric oxide.

## Follow up

After day 14, additional clinical data, such as time on ventilator support (days) and the use of drugs for the treatment of PH, are collected to answer the secondary outcome questions. Also, echocardiographic measurements are taken at 6 and 12 months to evaluate the presence of chronic PH (table 1).

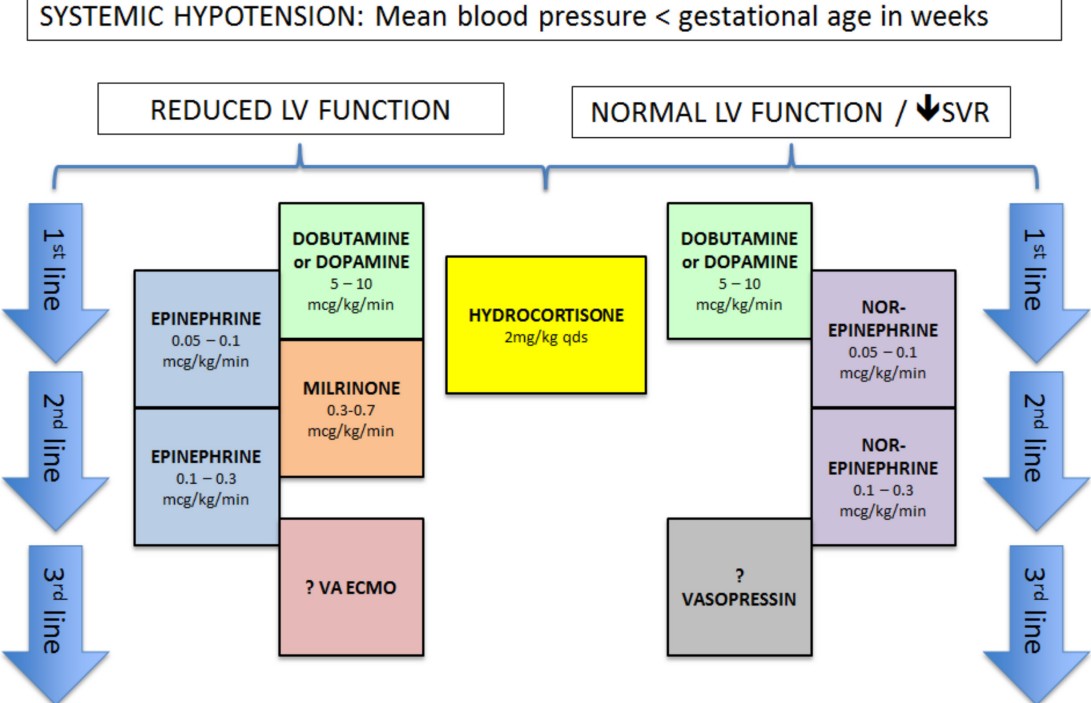

**Figure 3** Treatment flow chart of systemic hypotension. Flow chart that is added to the treatment protocol, showing the treatment plan for systemic hypotension. LV, left ventricular; SVR, systemic vascular resistance; VA ECMO, veno-arterial extracorporeal membrane oxygenation; ?, if available.

**Table 1** Procedures and measurements

| | Day 0–7 before start therapy | 3 hours after start sildenafil | 12 hours after start | 8 am after start | 24 hours after start | Day of surgery, preoperatively | Day after surgery | Day of ECMO, pre-cannulation | 8 am after start ECMO | Day 14 | Day 28/before discharge | Day 56 | 6 months | 12 months |
|---|---|---|---|---|---|---|---|---|---|---|---|---|---|---|
| Echocardiography | X | | | | X | | | | | X | X | X | X | X |
| Calculation OI | X | | X | | X | | | | | | | | | |
| Calculation VIS score | X | | X | | X | | | | | | | | | |
| Blood sample | X | | | X | | X | X | X | X | X | | | | |
| Tracheal aspirate | X | | | X | | X | X | X | X | X | | | | |
| Urine sample | X | | | X | | X | X | X | X | X | | | | |
| Severity of CLD | | | | | | | | | | | X | | | |
| Ultrasound brain | X | | | | | | X | | | | | | | |
| Sildenafil plasma level | | X | | X | | X | X | | X | | | | | |

CLD, chronic lung disease; ECMO, extracorporeal membrane oxygenation; OI, oxygenation index; VIS, vasoactive-inotropic support.

## Data collection

Echocardiography parameters are measured by local physicians, centrally collected and reviewed by two blinded independent physicians to reduce inter-observer variation. Demographic and neonatal characteristics as well as data on the clinical course of all patients are entered in a password protected web-based database in Rotterdam (OpenClinica). On request the collected data will be available. All centres will keep a logbook of the number of non-participants, including the reasons for not participating. Study documents are securely stored at each study site for 15 years.

## Laboratory testing

Blood, urine and tracheal samples are collected in most centres during the trial. Blood samples are collected before the start of the study and at different time points until day 14. Some samples will be used to externally validate a NONlinear Mixed Effects Modeling (NONMEM) prediction model for sildenafil. The other samples will be used in future research on biomarkers to predict severity and outcome of PH in CDH patients. The samples are centrifuged for 6 min at 3000 rpm.[22] Thereafter, the plasma is removed and stored at −20°C or colder. The total amount of blood taken is maximal 2.5% of the circulating volume. Blood sampling will only be done if a central or peripheral line is still present and/or in combination with routine laboratory measurements. This way blood sampling is a minimal burden for the patient.

Tracheal aspirate for proteomic analysis is also collected at different time points during routine tracheal suctioning in ventilated patients. Protein profiling with proteomics is used to identify specific groups of proteins that are involved in the pathogenesis of PH. The tracheal aspirates is centrifuged for 6 min at 3000 rpm and stored at −80°C.[23]

Also, 8 hour urine is collected at different time points. Two samples of 5 mL are taken and stored at −20°C or colder.

## Withdrawal of participants

Parents may decide to withdraw from the study at any time without any consequences. The investigator can decide to withdraw a patient from the study for urgent medical reasons. In some cases, there may be exclusion criteria, which were not known before randomisation. If this is the case, the patient will be withdrawn from the study after contacting the study coordinator. With consent of the parents data will still be collected, stored and analysed to perform an intention-to-treat analysis. These children will be treated according to standard practice.[10 16]

## Sample size calculation

The sample size calculation is based on a power analysis for the primary outcome, using previously published data on PH. Lusk *et al* showed that PH, defined as >2/3 systemic blood pressure measured on echocardiography, in CDH patients on day 14 has a positive predictive value

of 0.8 for death, death or ventilation, and death or ventilator support. PH on day 14 is observed in 64% of CDH patients.[24]

Even though the definition of the primary outcome is not the same, we assume a similar outcome percentage of 64% for failing the primary outcome in our trial, the absence of PH on day 14 without pulmonary vasodilator therapy and/or absence of death within the first 28 days of life, in the iNO group. Our aim is to promote practice change; therefore, we aim for a clinically significant difference. For a 25% relative reduction to 48%, a sample size of 300 patients (150 patients per group) is needed to obtain a power of 80%. This will match a number needed to treat of 6.25. Taking missing data and the effects of correction for covariates into account, we adjust this sample size to 330 patients. In the collaborating centres 550 patients will be born in 3 years. Based on our earlier trial (CMV Versus HFO in Congenital Diaphragmatic Hernia; VICI trial) we expect to have an inclusion rate of 60%. Therefore, the inclusion of 330 patients should be reached in 3 years.

### Data analysis

The patients will be analysed according to the group they are randomised to (intention-to-treat analysis). A p-value (two-sided) <0.05 is considered significant in all analyses. The primary endpoint will be analysed using multiple logistic regression with randomisation arm, centre, observed/expected head-lung ratio, position of the liver, side of the defect, defect size and ventilation modality as independent variables.[25] If necessary, multiple imputation using the fully conditional specification method will be used to account for missing data in the independent variables. We will perform a sensitivity analyses with adjustment for the use of prostin and milrinone, to account for the effects of these vasodilators on PH.

The following analyses will be performed for the secondary outcomes. The distribution of VIS score in all study participants will be compared between t=0 and t=12 hours after initiation of drug administration using a Wilcoxon signed rank test. The distribution of changes in OI and VIS score from t=0 to t=12 and t=24 hours will be compared between the randomisation groups with a Mann-Whitney test. The overall mortality in the first year of life will be compared between the randomisation groups with Kaplan-Meier curves and the log-rank test. The number of treatment failures, the need for ECMO (in ECMO centres) and the need for medication for PH or chronic lung disease at discharge, and during the first year of life, will be compared between randomisation groups with $\chi^2$ tests. The number of study drug free days at day 14, the number of ventilation-free days until day 28, the fraction of days with need for medical treatment (excluding the study drug) for PH during the hospital admission and the severity of chronic lung disease using the Bancalari definition will be compared between randomisation groups using Mann-Whitney tests. Deaths will be counted as the worst outcome in these analyses,

in accordance with the intention-to-treat principle. The presence of PH at 28 days/discharge, 6 and 12 months according to the echocardiographic parameters will be compared between randomisation groups with a $\chi^2$ test.

To externally validate the pharmacokinetic model of sildenafil and its active metabolite (in NONMEM) normalised prediction distribution errors and visual predictive check will be used. Furthermore, the model will be used to predict the drug concentrations from the new data set using simulations, in which we expect that the difference will be less than 20%. To assess whether there is a relationship between the concentration of sildenafil, its active metabolite and the clinical effects, such as OI, VIS score and echocardiography measures, a Mann-Whitney or Student's t-test will be used.

### Safety reporting and trial oversight

All severe adverse events (SAEs) and suspected unexpected serious adverse reactions (SUSARs) are reported from the enrolment until 12-month follow-up. Persistent or significant disability or incapacity that was not expected with the given observed to expected lung to head ratio (O/E LHR) is evaluated as an SAE. An elective hospital admission is not a SAE. All SAEs and SUSARs are reported to the approving ethics committees in accordance with their requirements. We will report the SAEs and SUSARs that result in death or are life threatening within 7 days of first knowledge. All other SAEs and SUSARs will be reported within a period of maximum 15 days. Once a year throughout the clinical trial, we will submit a safety report to the approving ethics committees and competent authorities of the countries involved.

The trial will be monitored by qualified, independent monitors. The trial is classified as a trial with moderate risk and a specific monitoring plan is in place.

The data safety monitoring board will monitor the incidence of mortality on a continuous basis. If at some point a large difference in mortality, defined as an absolute risk increase of 25%, between the two treatment groups is noticed, the data safety monitoring board may recommend ending the study.

Insurance will cover compensation to patients who suffer harm from trial participation.

## ETHICS AND DISSEMINATION

The trial will be submitted to the regulatory bodies and the local institutional review boards (IRB) in all participating countries. Important amendments will be communicated to all relevant parties. The study will be conducted according to the principles of the Declaration of Helsinki, in accordance with the Medical Research Involving Human Subjects Act and national rules and regulations on personal data protection. Parental informed consent will be obtained. The results of this study will be disseminated via peer-reviewed publications and implemented in the international guidelines for the treatment of newborns with CDH.

## Author affiliations
[1] Department of Intensive care and Pediatric Surgery, Erasmus University Rotterdam, Rotterdam, The Netherlands
[2] Department of Neonatology, University Medical Center, Mannheim, Mannheim, Germany
[3] Department of Women and Children's Health, School of Life Course Sciences, Faculty of Life Sciences and Medicine, King's College London, London, UK
[4] Department of Pediatrics, Division of Neonatology, Radboudumc Amalia Children's Hospital, Nijmegen, The Netherlands
[5] Department of Neonatology, Royal Hospital for Children Glasgow, Glasgow, UK
[6] Department of Development and Regeneration, KU Leuven, Leuven, Belgium
[7] Department of Biostatistics, Erasmus MC, Rotterdam, The Netherlands

**Collaborators** CDH Euro Consortium: Germany: Florian Kipfmueller, Department of Neonatology and Pediatric Critical Care Medicine, University Children's Hospital, Bonn. Spain: Maria Dolores Elorza, Ana Sanchez, Neonatology Department, Leopoldo Martinez, Pediatric Surgery Department, Carlos Labrandero, Viviana Arreo, Pediatric Cardiology Division, Hospital Universitario La Paz, Madrid. Africa: Pertierra Cortada, Jordi Clotet Caba Neonatology Department, Hospital Sant Joan de Déu Barcelona. Marta Aguar, Ana Gimeno, Raquel Escrig, Division of Neonatology, University and Polytechnic Hospital La Fe Valencia. Italy: Irma Capolupo, Pietro Bagolan, Department of Medical and Surgical Neonatology, Bambino Gesu' Children's Hospital, Rome. Fabrizio Ciralli, Genny Raffaeli, Giacomo Cavallaro, Valentina Condò, Fondazione IRCCS Ca' Granda Ospedale Maggiore Policlinico, NICU, University of Milan, Department of Clinical Sciences and Community Health. United Kingdom: United Kingdom - Paul D. Losty Department of Paediatric Surgery, Division of Child Health, Alder Hey Children's Hospital NHS Foundation Trust, University of Liverpool, Marie Horan, Paediatric Intensive Care Alder Hey Children's Hospital NHS Foundation Trust, University of Liverpool. Nimish V. Subhedar, NICU, Liverpool Women's Hospital, Liverpool. Yogen Singh, Department of Neonatology, Cambridge University Hospitals NHS Foundation trust, Cambridge. Emma E. Williams, The Asthma UK Centre in Allergic Mechanisms of Asthma; Women and Children's Health, School of Life Course Sciences, Faculty of Life Sciences and Medicine, King's College London, Denmark Hill, London. Theodore Dassios, Ravindra Bhat, King's College Hospital NHS Foundation Trust, London. Austria: Jennifer B. Brandt, Alexandra Kreissl, Angelika Berger, Department of Pediatrics and Adolescent Medicine, Division of Neonatology, Pediatric Intensive Care Medicine and Neuropediatrics, Medical University of Vienna. Berndt Urlesberger, Division of Neonatology, Department of Pediatrics and Adolescent Medicine, Medical University of Graz. Sweden: Carmen Mesas Burgos, Björn Frenckner, Department of Pediatric Surgery, Björn Larrson, Pediatric Intensive Care Unit, Karolinska University Hospital, Stockholm. Portugal: Carla Pinto, Serviço de Cuidados Intensivos Pediátricos, Hospital Pediátrico, Centro Hospitalar e Universitário de Coimbra, Coimbra. Joana Saldaha, Department of Neonatology, Hospital de Santa Maria, Lisbon. Belgium: Anne Debeer, Anne Smits, Neonatology, University Hospitals Leuven, Leuven. Norway: Ragnhild Emblem, Department of Pediatric Surgery, Oslo University Hospital, Oslo. Canada: Richard Keijzer, Department of Surgery, Yassar Elsayed, Department of Neonatology, Pediatrics and Child Health, University of Manitoba and Children's Hospital Research Institute of Manitoba. Australia: David Tingay, Department of Neonatology, Royal Children's Hospital, Melbourne, Australia. The Netherlands: Ulrike Kraemer, Intensive Care and department of Pediatric Surgery, Erasmus MC, Rotterdam.

**Contributors** All investigators of the Consortium described below have contributed to the design of the trial protocol and have approved this version for submission. Coordinating investigator SC-dO and DT are responsible for all aspects of the study conduct, practically study oversight, recruitment, training of the participating hospitals, reporting of the severe adverse events and suspected unexpected serious adverse reactions, outcome assessment and data management. DT, KA, TS, AvH, AG and NP are responsible for study oversight. JvR has contributed to statistical methods and will be involved in interpretation of the results. SC-dO will lead the dissemination and translation of results with the contribution of all investigators of the CDH EURO Consortium. Also all members will have authority over the data.

**Funding** This work was supported by CDH UK Sparks grant number 16EMC01 and by Stichting Sophia Kinderziekenhuis Fonds grant number S17-19.

**Competing interests** None declared.

**Patient consent for publication** Not required.

**Provenance and peer review** Not commissioned; externally peer reviewed.

## ORCID iDs
Suzan Cochius-den Otter http://orcid.org/0000-0002-6325-285X
Anne Greenough http://orcid.org/0000-0002-8672-5349

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
