## [Reviewer comments · BMJ Open]

ARTICLE DETAILS

TITLE (PROVISIONAL)	The CoDiNOS trial protocol: An international randomized controlled trial of intravenous sildenafil versus inhaled nitric oxide for the treatment of pulmonary hypertension in neonates with congenital diaphragmatic hernia.
AUTHORS	Cochius - den Otter, Suzan (proxy) (contact); Schaible, Thomas; Greenough, Anne; van Heijst, Arno; Patel, Neil; Allegaert, Karel; van Rosmalen, Joost; Tibboel, Dick

VERSION 1 – REVIEW

REVIEWER	Dr. Lauren E Kelly University of Manitoba, Canada No financial interests to declare. I have published previously with Dr. Allegaert.
REVIEW RETURNED	27-Jun-2019

GENERAL COMMENTS	The CoDiNOS trial protocol: An international randomized controlled trial of intravenous sildenafil versus inhaled nitric oxide for the treatment of pulmonary hypertension in neonates with congenital diaphragmatic hernia. Reviewer comments: This is a clinical trial protocol for an international RCT of IV sildenafil for pulmonary hypertension in neonates with congenital diaphragmatic hernia compared with the current first line treatment, nitric oxide. This is an appropriately designed open label study given the differences in administration route with blinded echocardiography analysis. The control arm is justified as nitric oxide is the current standard of care in the recruitment centres. The protocol is well written, detailed and follows the SPIRIT checklist guidelines for RCT protocol reporting. The collection of routine samples to validate a NONMEN prediction model for sildenafil is commendable and a valuable integration into this study design. I have a few comments to further strengthen the interpretation of this manuscript, and I recommend publication. Specific comments: Abstract List countries of recruitment or even Canada, Europe and Australia. (this is important to justify the control arm as iNO is not standard of care globally). The hypothesis is clear with MCD of 25% reduction in PH on day 14 – but is this just PH or the whole composite outcomes of PH +/- death first 28 days +/- vasodilators?
--

	Introduction Quantify the problem and population size, how many neonates are born with CDH? What are the risk factors for CDH +/- PH Could add to challenges with iNO, a discussion on availability in resource poor settings and not all neonates respond to iNO. The hypothesis as written "should be the first line of treatment for PH" implies to me that the implementation of this will be evaluated in this trial. Methods How will allocation be concealed? The rationale for the choice of outcome measures is missing – were parents (CDH UK Sparks) consulted on what was important to measure? How were these outcomes selected? Who is informing the parents about the study following the antenatal diagnosis? What is the block size for randomization? How many centres have ECMO available? How variable are the rescue protocols across centres? The MCD of 25% requires justification – would this reduction encourage centres to switch to sildenafil? How was this determined? How will your analysis be corrected for multiple comparisons? What are the stopping rules for the DSMB? What is a "large" difference in mortality? This should be prespecified. Will the DSMB be monitoring recruitment rates at each site? Who is monitoring the study? Recommend a risk-based approach. Are there potential challenges expected? Is there a regulatory approved sildenafil product for use in neonates in each country? If no, will a clinical trial application/waiver from the regulators be obtained?
--	---

REVIEWER	Pramod Puligandla Montreal Children's Hospital Division of Pediatric General and Thoracic Surgery Montreal, Quebec, Canada
REVIEW RETURNED	12-Jul-2019

GENERAL COMMENTS	This is a well designed, second RCT from the CDH Euroconsortium. It will be able to answer the specific question of the safety and efficacy of iv sildenafil as a primary treatment for pulmonary hypertension associated with CDH. The rationale is sound but the authors may benefit from specialized statistical support/assessment to ensure that the right tests are being performed. I have 2 comments:  1. Can the authors clarify how other pulmonary vasodilators will be handled during the study period (e.g. bosentan, prostacyclin)? Are these prohibited or just documented? How will the effect of these additional pulmonary vasodilators be accounted for? 2. the diagnosis of PH includes a pre- vs post-saturation difference of >10%. Is this too small a difference to accurately predict PH? Should this value be more like 20% difference?
--

VERSION 1 – AUTHOR RESPONSE

Reviewer: 1

Reviewer Name: Dr. Lauren E Kelly

Institution and Country: University of Manitoba, Canada

Please state any competing interests or state 'None declared': No financial interests to declare. I have published previously with Dr. Allegaert.

The CoDiNOS trial protocol: An international randomized controlled trial of intravenous sildenafil versus inhaled nitric oxide for the treatment of pulmonary hypertension in neonates with congenital diaphragmatic hernia.

Reviewer comments:

This is a clinical trial protocol for an international RCT of IV sildenafil for pulmonary hypertension in neonates with congenital diaphragmatic hernia compared with the current first line treatment, nitric oxide. This is an appropriately designed open label study given the differences in administration route with blinded echocardiography analysis. The control arm is justified as nitric oxide is the current standard of care in the recruitment centres. The protocol is well written, detailed and follows the SPIRIT checklist guidelines for RCT protocol reporting. The collection of routine samples to validate a NONMEN prediction model for sildenafil is commendable and a valuable integration into this study design. I have a few comments to further strengthen the interpretation of this manuscript, and I recommend publication.

Thank you for your kind comments and additional questions. Please find the answers below.

Specific comments:

Abstract

List countries of recruitment or even Canada, Europe and Australia. (this is important to justify the control arm as iNO is not standard of care globally).

We have added this to the abstract

The hypothesis is clear with MCD of 25% reduction in PH on day 14 – but is this just PH or the whole composite outcomes of PH +/- death first 28 days +/- vasodilators?

The 25% reduction accounts for the whole definition of the primary outcome; PH +/- death first 28 days +/- vasodilators. We have clarified this in the sample size calculation on page 11; Even though the definition is not the same, we assume a similar outcome percentage of 64% for failing the primary outcome in our trial, the absence of PH on day 14 without pulmonary vasodilator therapy and/or absence of death within the first 28 days of life, in the iNO group. For a 25% relative reduction to 48%, a sample size of 300 patients (150 patients per group) is needed to obtain a power of 80%.

Introduction

Quantify the problem and population size, how many neonates are born with CDH?

What are the risk factors for CDH +/- PH

We changed the introduction on page 4 to: Congenital diaphragmatic hernia (CDH) is a severe developmental defect of the diaphragm with an incidence of approximately 1 in 3000 live births and a mortality of 27%

On page 4 we have added: "Although the presence of lung hypoplasia can be predicted with prenatal parameters, reliable predictors for pulmonary hypertension in CDH patients are lacking [1]."

Could add to challenges with iNO, a discussion on availability in resource poor settings and not all neonates respond to iNO.

We changed the introduction on page 4 to: “However, in the two available randomized controlled trials (RCT) with a small number of patients with CDH, survival did not improve and more ECMO treatment was needed despite short-term improved oxygenation in the iNO treated group [2, 3]. In the centers of the CDH EURO Consortium, iNO is standard of care in infants with CDH and PH although the positive pharmacodynamic effects in these infants are less convincing than in infants with PPHN [4, 5]. The pathophysiological mechanism of this difference is not understood. In resource poor settings iNO is often unavailable. In the surge to find another treatment option, trials to evaluate the effect of sildenafil in newborns with PPHN have been conducted [6].”

The hypothesis as written “should be the first line of treatment for PH” implies to me that the implementation of this will be evaluated in this trial.

We will not evaluate the implementation of sildenafil as first line of treatment in this trial, we first test the hypothesis that sildenafil is superior to iNO. This will be clear at the end of the trial or earlier as the DSMB can stop the trial if the difference between the two treatment arms is over 25%. However, if sildenafil is superior, we will implement this in the CDH-Euro Consortium guidelines. As sildenafil has been tested in resource poor settings, we do not expect major problems with the implementation of sildenafil as treatment option.

Methods

How will allocation be concealed?

Page 8: When the patient meets the inclusion criteria, the physician logs in to the web based program which randomizes the patient with a computer-generated randomization list, made by the independent statistician of the Data Safety and Monitoring Board.

The rationale for the choice of outcome measures is missing – were parents (CDH UK Sparks) consulted on what was important to measure? How were these outcomes selected?

CDH UK Sparks did not advise on the outcome measures. However, they did review the protocol and agreed to its content. As discussed on page 11, we selected pulmonary hypertension on day 14 as primary outcome as this predicts death, death or ventilation, and death or ventilator support. Long-term ventilator support causes substantial morbidity and prolongs hospital stay.

Who is informing the parents about the study following the antenatal diagnosis?

We clarified this in the text on page 7: Following antenatal diagnosis, the parents are counselled and informed about the study by the clinician or research coordinator, leaving them ample time to consider parental approval during the course of the pregnancy.

What is the block size for randomization?

To reduce the predictability of the randomized treatment assignments and the associated selection bias, the investigators remain masked to the block size, and the block sizes are thus not presented in the manuscript. Also block sizes vary randomly[7].

How many centres have ECMO available? How variable are the rescue protocols across centres?

Fourteen centers use ECMO in CDH patients. The rescue protocol is similar in the centers as treatment is per CDH Euro Consortium protocol. The only difference is the use of ECMO in the different centers. We have further standardized care by adding an inotropic support flow chart to the protocol. (page 9 and figure 3))

The MCD of 25% requires justification – would this reduction encourage centres to switch to sildenafil? How was this determined?

The goal of the trial is to substantially increase outcome in CDH patients. In encephalopathic newborns treated with therapeutic hypothermia, the number of patients who need to be treated to prevent one infant from dying or being disabled was 6 to 7[8]. We consider this a substantial improvement, which has changed the standard of care for these patients. Therefore, we used this as a goal for our trial.

How will your analysis be corrected for multiple comparisons?

The different analyses represent different research questions and have different outcome measures. Therefore we will consider and discuss each test and p-value separately, and we think that no correction for multiple testing is needed to account for the number of outcome measures. The independent variables in the logistic regression (except randomization arm) are confounders to better estimate the treatment efficacy, and no adjustment for multiple testing is needed there.

What are the stopping rules for the DSMB? What is a “large” difference in mortality? This should be prespecified. Will the DSMB be monitoring recruitment rates at each site?

We defined a large difference in mortality an absolute risk increase of 25%. The DSMB will also monitor recruitment rates at each site, although we expect big differences in recruitment rate between centers as CDH is an orphan disease and some centers treat 5 patients per year, others over 40.

Who is monitoring the study? Recommend a risk-based approach.

The trial will be monitored by qualified, independent monitors. The trial is classified as a trial with moderate risk and a specific monitoring plan is in place.

Are there potential challenges expected?

The biggest changes is the approval of the trial by the IRB and National Drug Authority in the different countries due to the many rules that are in place for drug trials, especially in children. This has already taken a lot of time and has so far slowed us down in starting the trial in most of the centers.

Is there a regulatory approved sildenafil product for use in neonates in each country? If no, will a clinical trial application/waiver from the regulators be obtained?

Intravenous sildenafil and inhaled nitric oxide are both regulatory approved available in the countries involved.

Reviewer: 2

Reviewer Name: Pramod Puligandla

Institution and Country: Montreal Children's Hospital, Division of Pediatric General and Thoracic Surgery, Montreal, Quebec, Canada

Please state any competing interests or state 'None declared': None declared

This is a well-designed, second RCT from the CDH Euroconsortium. It will be able to answer the specific question of the safety and efficacy of iv sildenafil as a primary treatment for pulmonary hypertension associated with CDH. The rationale is sound but the authors may benefit from specialized statistical support/assessment to ensure that the right tests are being performed.

Thank you for your kind comments. A statistician, Joost van Rosmalen, PhD, working at the Department of Biostatistics of the Erasmus MC, has been involved in the development of the trial.

I have 2 comments:

1. Can the authors clarify how other pulmonary vasodilators will be handled during the study period (e.g. bosentan, prostacyclin)? Are these prohibited or just documented? How will the effect of these additional pulmonary vasodilators be accounted for?

The use of bosentan as add on therapy is allowed and is considered as PH treatment on day 14. Milrinone and prostin use are allowed and will be documented in the CRF. Other pulmonary vasodilators are not allowed (page 9). We will correct for this using a sensitivity analyses with adjustment for milrinone and prostin (page 9 and 11).

2. the diagnosis of PH includes a pre- vs post-saturation difference of >10%. Is this too small a difference to accurately predict PH? Should this value be more like 20% difference?

The diagnosis of PH consist of two out of four criteria. Therefor pre- vs post-saturation difference of >10% is not the only sign of PH. The difference of 10% is used as a guideline for treatment of PH in the CDH Euro Consortium Consensus and is therefore chosen in the trial[9].

1. Russo, F.M., et al., Lung size and liver herniation predict need for extracorporeal membrane oxygenation but not pulmonary hypertension in isolated congenital diaphragmatic hernia: systematic review and meta-analysis. *Ultrasound Obstet Gynecol*, 2017. 49(6): p. 704-713.
2. Inhaled nitric oxide and hypoxic respiratory failure in infants with congenital diaphragmatic hernia. The Neonatal Inhaled Nitric Oxide Study Group (NINOS). *Pediatrics*, 1997. 99(6): p. 838-45.
3. Clark, R.H., et al., Low-dose nitric oxide therapy for persistent pulmonary hypertension of the newborn. Clinical Inhaled Nitric Oxide Research Group. *N Engl J Med*, 2000. 342(7): p. 469-74.
4. Reiss, I., et al., Standardized postnatal management of infants with congenital diaphragmatic hernia in Europe: the CDH EURO Consortium consensus. *Neonatology*, 2010. 98(4): p. 354-64.
5. Putnam, L.R., et al., Evaluation of Variability in Inhaled Nitric Oxide Use and Pulmonary Hypertension in Patients With Congenital Diaphragmatic Hernia. *JAMA Pediatr*, 2016.
6. Kelly, L.E., A. Ohlsson, and P.S. Shah, Sildenafil for pulmonary hypertension in neonates. *Cochrane Database Syst Rev*, 2017. 8: p. CD005494.
7. Rosenberger, W.F. and M.L. Lachin, *Randomization in Clinical Trials 2015*: John Wiley & Sons, Inc 254.
8. Tagin, M.A., et al., Hypothermia for neonatal hypoxic ischemic encephalopathy: an updated systematic review and meta-analysis. *Arch Pediatr Adolesc Med*, 2012. 166(6): p. 558-66.
9. Snoek, K.G., et al., Standardized Postnatal Management of Infants with Congenital Diaphragmatic Hernia in Europe: The CDH EURO Consortium Consensus - 2015 Update. *Neonatology*, 2016. 110(1): p. 66-74.

VERSION 2 – REVIEW

REVIEWER	Dr. Lauren Kelly University of Manitoba, Canada I have published with one of the co-authors (KA) previously in a different disease area.
REVIEW RETURNED	29-Aug-2019

GENERAL COMMENTS	This is a clinical trial protocol for an international RCT of IV sildenafil for pulmonary hypertension in neonates with congenital diaphragmatic hernia compared with the current first line treatment, nitric oxide. This is an appropriately designed open label study given the differences in administration route with blinded echocardiography analysis. The control arm is justified as nitric oxide is the current standard of care in the recruitment centres. The protocol is well written, detailed and follows the SPIRIT checklist guidelines for RCT protocol reporting. The collection of routine samples to validate a NONMEN prediction model for sildenafil is commendable and a valuable integration into this study design. The authors have addressed most of my previous comments including addressing the limitations of iNO in resource poor settings, clinical trial monitoring and modified hypothesis. There are some minor revisions requested including clarifications of the method for allocation concealment (missing from Page 36) and minor spelling corrections (e.g. dough vs though (page 39 line 20). The stopping rules needs a prespecified definition of a "large"
---

	difference in mortality. It is unclear if this trial will be submitted to regulatory bodies (e.g. a Clinical trial application with Health Canada) as this is an off-label use of sildenafil. Finally, it is still unclear to me how/why/if a 25% reduction in PH is meaningful primary endpoint. Would this promote practice change? How was 25% chosen? Elaboration here would be helpful for interpretation.
--	---

REVIEWER	Pramod Puligandla Montreal Children's Hospital Division of Pediatric General and Thoracic Surgery Montreal, Quebec, Canada
REVIEW RETURNED	15-Aug-2019

GENERAL COMMENTS	The authors have addressed the concerns raised after initial review. I feel that the paper is now fit for publication.
--

VERSION 2 – AUTHOR RESPONSE

Reviewer: 2

Reviewer Name: Pramod Puligandla

Institution and Country: Montreal Children's Hospital Division of Pediatric General and Thoracic Surgery Montreal, Quebec, Canada Please state any competing interests or state 'None declared':

None declared

The authors have addressed the concerns raised after initial review. I feel that the paper is now fit for publication.

Thank you for your positive evaluation.

Reviewer: 1

Reviewer Name: Dr. Lauren Kelly

Institution and Country: University of Manitoba, Canada

Please state any competing interests or state 'None declared': I have published with one of the co-authors (KA) previously in a different disease area.

This is a clinical trial protocol for an international RCT of IV sildenafil for pulmonary hypertension in neonates with congenital diaphragmatic hernia compared with the current first line treatment, nitric oxide. This is an appropriately designed open label study given the differences in administration route with blinded echocardiography analysis. The control arm is justified as nitric oxide is the current standard of care in the recruitment centers. The protocol is well written, detailed and follows the SPIRIT checklist guidelines for RCT protocol reporting. The collection of routine samples to validate a NONMEN prediction model for sildenafil is commendable and a valuable integration into this study design. The authors have addressed most of my previous comments including addressing the limitations of iNO in resource poor settings, clinical trial monitoring and modified hypothesis.

Thank you for your positive feedback.

There are some minor revisions requested including clarifications of the method for allocation concealment (missing from Page 36)

We have added to page 8: Participants will be randomized using ALEA, which is an online, central randomization service (<https://www.aleaclinical.eu>) . Allocation concealment will be ensured, as the service will not release the randomization code until the patient has been recruited into the trial, which takes place after all baseline characteristics have been added.

and minor spelling corrections (e.g. dough vs though (page 39 line 20)).

We have changed dough to though on page 11.

The stopping rules needs a prespecified definition of a "large" difference in mortality.

We defined a large difference in mortality an absolute risk increase of 25%. We have added this on page 12 to "Safety reporting and trial oversight".

It is unclear if this trial will be submitted to regulatory bodies (e.g. a Clinical trial application with Health Canada) as this is an off-label use of sildenafil.

Yes, you are correct. As this is an off-label use of sildenafil in a pediatric population, the trial will be submitted to the regulatory bodies and the local IRB's in all participating countries. We have added that to "Ethics and dissemination" on page 13.

Finally, it is still unclear to me how/why/if a 25% reduction in PH is meaningful primary endpoint. Would this promote practice change? How was 25% chosen? Elaboration here would be helpful for interpretation.

We are sorry we haven't answered the question more clear in our previous response. The goal of the trial is to substantially increase outcome in CDH patients. There are no trials comparable to our trial in CDH patients. In an attempt to refer other type of interventions in term neonates, encephalopathic newborns treated with therapeutic hypothermia, the number of patients who need to be treated to prevent one infant from dying or being disabled was 6 to 7 [1]. We consider this a substantial improvement, which has changed the standard of care for these patients [2]. Therefore, we used this as a goal for our trial. A reduction of 25% matched a number needed to treat of 6.25 patients. As pulmonary hypertension is a key pathological finding and determinant of disease severity in CDH, we considered that reducing the proportion of infants with severe PH by 25% would represent a clinically meaningful improvement in acute clinical status, demonstrating a real response, or not, to therapy. We have made our goal more clear in the "sample size calculation" section on page 11: Even though the definition is not the same, we assume a similar outcome percentage of 64% for failing the primary outcome in our trial, the absence of PH on day 14 without pulmonary vasodilator therapy and/or absence of death within the first 28 days of life, in the iNO group. Our aim is to promote practice change, therefore we aim for a clinically significant difference. For a 25% relative reduction to 48%, a sample size of 300 patients (150 patients per group) is needed to obtain a power of 80%. This will match a number needed to treat of 6.25.

1. Tagin, M.A., et al., Hypothermia for neonatal hypoxic ischemic encephalopathy: an updated systematic review and meta-analysis. Arch Pediatr Adolesc Med, 2012. 166(6): p. 558-66.
2. <https://www.nice.org.uk/guidance/ipg347/documents/therapeutic-hypothermia-with-intracorporeal-temperature-monitoring-for-hypoxic-perinatal-brain-injury-interventional-procedures-overview2>. 2010.

VERSION 3 – REVIEW

REVIEWER	Lauren Kelly The University of Manitoba, Canada Have published previously with KA in a different disease area. No competing interests.
REVIEW RETURNED	18-Sep-2019
GENERAL COMMENTS	All my previous comments have been addressed.